# Anti-Inflammatory Activity of No-Ozone Cold Plasma in *Porphyromonas gingivalis* Lipopolysaccharide-Induced Periodontitis Rats

**DOI:** 10.3390/ijms25116161

**Published:** 2024-06-03

**Authors:** Kwang-Ha Park, Yoon-Seo Jang, Ji-Young Joo, Gyoo-Cheon Kim, Jeong-Hae Choi

**Affiliations:** 1Corporate Affiliated Research Institute, Feagle Co., Ltd., 47, Soekgyesandan 2-gil, Sangbuk-myeon, Yangsan 50561, Republic of Korea; pkh900909@gmail.com; 2Department of Anatomy and Cell Biology, School of Dentistry, Pusan National University, Yangsan 50612, Republic of Korea; 3Department of Oral and Maxillofacial Surgery, School of Dentistry, Pusan National University, Yangsan 50612, Republic of Korea; 4Department of Periodontology of Dental Research Institute, Pusan National University Dental Hospital, Yangsan 50612, Republic of Korea

**Keywords:** no-ozone cold plasma, periodontitis, anti-inflammatory, electron

## Abstract

Periodontitis is an inflammatory disease caused by *Porphyromonas gingivalis* (*P. gingivalis*) in the oral cavity. This periodontal disease causes damage to the periodontal ligament and alveolar bone and can cause tooth loss, but there is no definite treatment yet. In this study, we investigated the possibility of using no-ozone cold plasma to safely treat periodontitis in the oral cavity. First, human gingival fibroblasts (HGFs) were treated with *P. gingivalis*-derived lipopolysaccharide (PG-LPS) to induce an inflammatory response, and then the anti-inflammatory effect of NCP was examined, and a study was conducted to identify the mechanism of action. Additionally, the anti-inflammatory effect of NCP was verified in rats that developed an inflammatory response similar to periodontitis. When NCP was applied to PG-LPS-treated HGFs, the activities of inflammatory proteins and cytokines were effectively inhibited. It was confirmed that the process of denaturing the medium by charged particles of NCP is essential for the anti-inflammatory effect of NCP. Also, it was confirmed that repeated treatment of periodontitis rats with NCP effectively reduced the inflammatory cells and osteoclast activity. As a result, this study suggests that NCP can be directly helpful in the treatment of periodontitis in the future.

## 1. Introduction

Periodontitis is a common dental and chronic inflammatory disease caused by periodontal tissue infection [1]. Periodontitis causes inflammatory reactions in the periodontal tissue and tooth root, characterized by periodontal ligament damage and alveolar bone loss. If left untreated, a periodontal pocket forms between the gum and the base of the root, causing the tooth to shake and lose shape. Periodontitis is caused by long-term exposure of the periodontal tissue to lipopolysaccharides (LPSs) present in the cell membrane of oral microorganisms, which attach to the tooth surface and form calculi. *Porphyromonas gingivalis* (*P. gingivalis*), *Tannerella forsythia* (*T. forsythia*), and *Treponema denticola* (*T. denticola*) are oral microorganisms that play major roles in inducing periodontitis [2]. These three microorganisms are oral bacteria mainly found in periodontitis patients and are collectively referred to as the “red complex” [3]. *T. forsythia* is an important mediator of biofilm formation, and *T. denticola* is known to affect bone resorption. Along with these oral bacteria, *P. gingivalis* is considered the main cause of chronic periodontitis. In clinical practice, many methods of treating periodontitis include scaling and administering antibiotics. They significantly affect the recovery of damaged periodontal tissue and alveolar bone, although they reduce pathogens that cause inflammation. Also, various devices are being developed for more effective removal of residual microorganisms after calculus and physical removal procedures that cause periodontitis. These mainly minimize the number of residual microorganisms by using high-temperature light sources (lasers and LED) in combination with physical removal and low-concentration hydrogen peroxide [4]. However, these technologies have a disadvantage in that oral tissues may suffer thermal damage because of high thermal energy [5,6]. Furthermore, even if oral microorganisms are sterilized, LPSs in the residue can cause inflammation, and light-source devices that use them have no impact on the pre-existing inflammatory response. Therefore, developing a new technology capable of effectively removing the causative bacteria of periodontitis and suppressing inflammatory reactions is necessary.

Plasma is an ionized active gas generated when high energy is applied to the gas, and it represents the fourth state of matter. In generating plasma, electrons; ions; physical elements, such as thermal energy, UV, and electric fields; and various chemical factors, such as reactive nitrogen and oxygen species, are generated. Recently, low-temperature plasma at temperatures similar to body temperatures has been shown to sterilize microorganisms [7,8,9], have anti-inflammatory reactions [10], have anticancer effects [11,12], promote skin regeneration (e.g., wound healing) [13], and cure skin diseases such as atopic dermatitis [14]. Research is being conducted to introduce various medical effects, such as therapeutic effects and the promotion of the transdermal absorption of drugs [15,16,17]. The most common medical effect of plasma is a sterilizing effect, and research on the sterilizing effect on numerous bacteria existing in the oral cavity, including *P. gingivalis*, has been introduced and actively conducted as a new method for killing pathogens in the oral cavity [18]. However, in order to apply the oral microbial killing effects of low-temperature plasma for treatment, the problem of ozone generated by the plasma must be solved. Ozone contributes to the sterilization effect of plasma; however, continuous exposure to ozone can cause respiratory problems, such as asthma, dyspnea, and airway inflammation. Therefore, the World Health Organization (WHO) has limited the acceptable standard of ozone in the air to 0.05 ppm [19,20].

A previous study reported that ozone at a level below the WHO standard and temperature of <30 °C (no-ozone cold plasma, NCP) effectively killed various oral microorganisms, including *P. gingivalis* [21]. However, the effects of NCP on various inflammatory reactions in periodontitis have not been investigated. Herein, we investigated the effects of NCP on the inflammatory response in periodontitis. PG-LPS, an LPS derived from the wall of *P. gingivalis*, was used to induce the same inflammatory response as periodontitis in HGFs and an animal model. Furthermore, another study was conducted to identify the mechanism of action of NCP in regulating the inflammatory response of periodontitis.

## 2. Results

### 2.1. NCP Treatment for Up to 5 Min Has No Effects on Cell Viability

To confirm the effect of NCP on the growth of periodontal cells, HGFs were treated with NCP, and cell viability was confirmed by an SRB assay. After 24 h of treating HGFs with NCP, no round-up or dead cells were observed. Upon analyzing the change in cell viability by measuring the absorbance, it was confirmed that there was no significant difference in either of the experimental groups compared with the control group (Figure 1B).

### 2.2. NCP Inhibited PG-LPS-Mediated Inflammatory Reactions of HGFs

To examine the effect of NCP on the inflammatory response, an experiment was performed to establish the condition for inducing an immune response in periodontitis when HGFs were treated with PG-LPS. Appendix A shows that the level of NF-kB, a protein that regulates inflammation, increased immediately after PG-LPS treatment for up to 1 h and gradually decreased after 3 h. It was confirmed that the expression of *IL-1β* and *IL-6* changed with the change in the activity of NF-kB.

Using the conditions described above, a study was conducted to determine how NCP affects the inflammatory response induced by PG-LPS. As shown in Figure 1C, *IL-1β* in cells treated with NCP was reduced by 25% for 1 min and 60% for 3 min and decreased similarly to the control group for 5 min. *IL-6* was equally reduced by 13% for 1 min and 32% for 3 min and no longer increased for 5 min. Since *IL-1β* and *IL-6* significantly decreased when treated with NCP for >3 min, it was confirmed that the anti-inflammatory effect of NCP on the inflammatory response caused by PG-LPS appeared when treated for >3 min.

Western blotting was performed to examine whether the changing activities of NF-kB and STAT3, which are inflammation-regulating proteins, were induced by PG-LPS. As shown in Figure 1D, it was confirmed that treatment with NCP effectively inhibited p-NF-kB and p-STAT3 induced by PG-LPS.

### 2.3. NCP-Mediated Changes in PG-LPS-Containing Media Are Essential for Anti-Inflammatory Activity

To determine whether NCP shows anti-inflammatory efficacy even when cells and media are treated separately, we divided the experimental method into three types. In addition to the existing direct processing method in which both cells and medium are present (DT), two methods were added. In order to minimize the impact of the NCP-treated medium, treatment was carried out using a direct treatment + medium change (DT+MC) method in which a new medium containing PG-LPS was replaced immediately after treatment, and an indirect treatment method in which only a cell-free PG-LPS medium was treated with NCP and then transferred to cells (IDT).

Figure 2B shows changes in *IL-1β* and *IL-6* after treatment with NCPs in three different ways. Two pro-inflammatory cytokines were effectively suppressed in the DT method. *IL-6* was more effectively suppressed in DT+MC than DT. However, DT was more effective than DT+MC in suppressing *IL-1β*. Additionally, *IL-1β* and *IL-6* in the cells treated by IDT were lower than those treated by DT. Western blotting was used to examine changes in the activity of NF-kB and STAT3 (Figure 2C). As a result, all three treatment methods effectively inhibited the phosphorylation of two proteins. However, DT+MC showed the lowest inhibitory effect.

### 2.4. Charged NCP Particles Were Responsible for NCP-Mediated Anti-Inflammatory Activity in the PG-LPS-Treated HGFs

To determine whether NCP factors affect anti-inflammatory properties, an experiment was conducted using two types of mesh. Metal mesh (EG) using copper will not allow the passage of charged elements such as electrons when connected to a power source. On the other hand, non-metallic mesh (DE) made of cloth allows all factors generated by plasma to pass through.

As shown in Figure 3B, when DE mesh was placed between cells and NCP device, IL-1β and IL-6 were effectively inhibited. In contrast, the anti-inflammatory effect of NCP was significantly reduced placing the EG mesh between the cells and device. Additionally, inhibition of p-NF-kB remained the same using DE mesh but slightly increased using EG mesh (Figure 3C).

### 2.5. NCP Reduces the Inflammatory Phenotype of PG-LPS-Induced Periodontitis in Rats

A study was conducted to confirm the anti-inflammatory effect of NCP in periodontitis-induced oral tissues. As shown in Figure 4C, the boundary between the first molar and the alveolar bone was not clear in the PG-LPS group compared to the control group. In the PG-LPS+NCP group, the boundary between the first molar and alveolar bone became clear. Furthermore, the micro-CT three-dimensional image confirmed the size of the hole between the first molar and alveolar bone increased significantly during PG-LPS treatment. However, the size decreased in the PG-LPS+NCP group. In H&E staining results (Figure 4D), the density of cells in the tissues in the tooth ligament was significantly increased in the PG-LPS group compared with the control group. However, the cell density was reduced in the PG-LPS+NCP group. In RT-PCR results, *IL-1β* and *TNFα* were increased 2.5 times and 1.6 times in the PG-LPS group, but reduced in the PG-LPS+NCP group (Figure 4E).

### 2.6. NCP Treatment Inhibited the Accumulation of T Cells and Macrophages and Blocked the Activation of Osteoclasts in the Lesion

We examined the effect of NCP treatment on PG-LPS-induced periodontitis-like tissues. As shown in Figure 5A, the number of T cells between the first and second molars was significantly increased in the PG-LPS group compared to the control group. However, in the PG-LPS+NCP group, the number of T cells was similar to that in the control group. This phenomenon was also observed in the results for confirming macrophages. Figure 5B shows a significantly increased number of macrophages observed among the tooth ligaments, tooth root, and alveolar bone in the PG-LPS group. However, the number of macrophages was significantly reduced in the PG-LPS+NCP group compared with that in the PG-LPS group. Subsequently, the effect of NCP treatment on osteoclasts in tissues induced with periodontitis by PG-LPS was confirmed. As shown in Figure 5C, osteoclast activity was extensively increased between the root of the first molar and alveolar bone in the PG-LPS group. However, osteoclast activity was decreased in the PG-LPS+NCP group.

## 3. Discussion

The most frequently used method to treat periodontitis is to physically remove the causative bacteria by removing calculus through scaling. Because residual microorganisms may exist after removal, antibiotics or devices using light sources, such as lasers or LEDs, are used. However, the downside of these methods is that they can lead to drug resistance, tissue damage, and additional inflammatory responses. Therefore, a new treatment method is needed to kill microorganisms effectively and alleviate inflammatory reactions.

In this study, we induced an inflammatory response with PG-LPS in cells and an animal model and then treated them with NCP to see if the inflammatory response was mitigated. In the in vitro experiments, HGFs were used because they play a role in the repair between the periodontium and the tooth root [22]. Before the full-scale research, the NCP device was used to confirm that the treatment of HGFs with NCP did not cause any cytotoxicity in the cells.

Based on the safety of NCP, we investigated whether the inflammatory response in periodontitis could be suppressed. According to Pan et al. and Ambili et al., when periodontal tissue is exposed to PG-LPS, gingival fibroblasts bring inflammatory cells such as T cells into nearby tissues by secreting pro-inflammatory cytokines, such as *IL-1β* and *IL-6*. Figure 1C indicated that NCP effectively suppressed periodontitis inflammation caused by PG-LPS [23,24]. According to Souza et al., increased pro-inflammatory cytokines during PG-LPS treatment are caused by inflammatory regulating proteins such as NF-kB and STAT-3 [25]. The results of this study suggest the possibility that NCP suppressed *IL-1β* and *IL-6* by inhibiting NF-kB and STAT-3 activation.

Herein, we identified the specific mechanism of action by which NCP inhibits the inflammatory response induced by PG-LPS in HGFs. The anti-inflammatory effect was the best in the IDT method, suggesting that the inflammatory response was inhibited by changing the composition of the medium by factors such as electric field, electrons, and light. The IDT method showed a better anti-inflammatory effect than the DT method, which suggests that some of the working elements of NCP directly stimulated the cells and inhibited the anti-inflammatory effect.

Based on the experimental results, we conducted a study to identify the important factors in NCP that worked to inhibit the inflammatory response. It was observed that during NCP treatment with DE mesh, the anti-inflammatory effect was similar to that of the DT method. However, when EG mesh was installed, the anti-inflammatory effect of NCP almost disappeared. This indicates that the impact of metal mesh and electrical grounding may remove charged particles such as electrons and ions generated by NCP, indicating that charged particles are an important factor in the anti-inflammatory effect of NCP. Connecting this finding with the results in Figure 2, it is suggested that the charged particles in NCP induce changes in the composition of the medium, stimulating the cells to exert anti-inflammatory effects.

Based on the in vitro studies, we induced periodontitis in an animal model and observed histologic changes to determine the effectiveness of NCP in treating periodontitis. Figure 4 results are the same as those observed when PG-LPS-treated HGFs were treated with NCP, which indicates that the effect of NCP on the inflammatory response of periodontitis appears the same as that in HGFs and oral tissue. According to Cheng et al., *IL-1β* overexpression is a phenomenon found in tissues of periodontitis patients, and *IL-1β* is involved in leukocyte recruitment and neutrophil infiltration [26]. Furthermore, since *IL-1β* strongly promotes bone resorption, it was introduced as a representative cytokine associated with the occurrence of periodontitis. Therefore, the strong inhibition of *IL-1β* by NCP suggests the possibility that repeated treatment with NCP can effectively inhibit the inflammatory response and bone resorption of periodontitis. The inhibition of *IL-1β* observed in periodontal tissues suggests that NCP can effectively inhibit T-cell recruitment to the PG-LPS injection site and reduce macrophages (Figure 5A,B).

Meanwhile, osteoclasts, called resident macrophages [27], have been reported to be involved in bone damage, such as in alveolar bone or joints, by inducing the differentiation and activation by macrophages and T cells in periodontitis [28,29,30]. Through the TRAP assay, increased osteoclast activity significantly decreased when NCP was administered. The results of this study indicate that NCP effectively inhibited the osteoclast activity induced by PG-LPS and that the anti-inflammatory effect of NCP led to the suppression of osteoclast activity. These results may be helpful for the recovery of damaged tissues by suppressing osteoclast activity.

In conclusion, NCP has an anti-inflammatory effect against periodontitis induced by PG-LPS in HGFs and osteoclast inhibition activity in an animal model. The strong anti-inflammatory efficacy of NCP against periodontitis confirmed that NCP could be effective in treating periodontitis.

## 4. Methods and Materials

### 4.1. Plasma Device

The NCP device used in this study formed a plasma plume by directly applying electrical energy to high-purity (99.9%) argon gas. A dielectric barrier discharge-type plasma source is installed in the handpiece part where the plasma is generated (Figure 1A). The end of the plasma-generating nozzle of the handpiece is designed to be equipped with various types of dental tips. When the device is operated, argon gas flows between the electrode and nozzle. A 3 kVpp output with a frequency of 20 kHz is applied to both ends of the electrodes to generate plasma. The argon gas flow rate in the handpiece can be adjusted using a control knob. The experiment was conducted at a flow rate of 1 L/min.

### 4.2. In Vitro Study

#### 4.2.1. Cells

HGFs were purchased from the American Type Culture Collection. Cells were cultured in an incubator at 37 °C and 5% CO_2_ in a culture medium containing 1% antibiotics and 10% fetal bovine serum (FBS).

#### 4.2.2. Sulforhodamine B (SRB) Cell Growth Assay

Before full-scale research was started, an experiment was conducted to confirm the safety of the NCP device for cells. HGFs were seeded in a 35 mm dish and cultured in an incubator at 37 °C and 5% CO_2_ for 24 h. In the NCP treatment group, the distance between the tip of the handpiece and the cells was maintained at 1 cm. NCP was applied for 1, 3, and 5 min. After 24 h of NCP treatment, the medium was removed, and 4% paraformaldehyde (PFA) was dispensed to fix the cells on the dish for 30 min at room temperature. Next, 4% PFA was removed, and the dish was washed with DW five times. The dish was dried before the addition of 0.4% SRB. The cells were stained for 10 min at room temperature. After 0.4% SRB solution was then removed, the dish was washed five times for 5 min with 1% acetic acid solution and dried. Next, 10 mM Tris solution was added to each dish and shaken to dissolve the SRB crystals stained on the cells; 100 μL of this solution was transferred to a 96-well plate, and the absorbance was measured by UV spectroscopy at 515 nm.

#### 4.2.3. Western Blot

HGFs were treated with PG-LPS at a concentration of 1 μg/mL, then treated with plasma for 1, 3, and 5 min, and finally cultured in an incubator for 2 h. After cell lysis using RIPA buffer, the proteins were separated by centrifugation at 12,000 rpm for 10 min. Quantification of the extracted proteins was performed using the Bradford assay. A sample containing 30 μg of protein was prepared, electrophoresed on SDS–polyacrylamide gel, separated according to size, and transferred to a polyvinylidene fluoride membrane (PVDF, Millipore, Burlington, MA, USA). The PVDF membrane was cut to fit the size of the antibody to be identified. The cut PVDF membrane was blocked with 5% skim milk solution for 1 h. The membrane was incubated with the primary antibody overnight at 4 °C, and HRP-conjugated IgG was used as the secondary antibody after washing. Protein expression levels were confirmed using Amersham Imager 680 (GE Healthcare, Chicago, IL, USA).

#### 4.2.4. RNA Extraction and RT-PCR

HGFs were treated with PG-LPS at a concentration of 1 μg/mL, treated with low-temperature plasma for 1, 3, and 5 min, and then cultured in an incubator for 2 h. After washing with DPBS, 500 μL of Trizol reagent (Invitrogen, Waltham, MA, USA) was added; then, the cells were collected with a scraper and placed into an Eppendorf tube. After 100 μL of chloroform was dispensed into each tube and centrifugation at 12,000 rpm for 10 min, 200 μL of the clear supernatant was transferred to a new tube, mixed with the same amount of isopropanol, and centrifuged at 12,000 rpm for 10 min. After the supernatant was removed, the pellet was mixed with 500 μL of 75% ethanol for washing and centrifuged at 12,000 rpm for 10 min. Following this, ethanol was removed, and the pellet was dried. When the pellet was dried, 50 μL of DEPC water was added, and total RNA was quantified by UV spectroscopy. RNA was synthesized into cDNA using Maxime RT Premix (Intron, Gyeonggi-do, Republic of Korea), followed by RT-PCR with denaturation at 95 °C for 30 s, annealing for 30 s, and extension at 72 °C for 60 s. The primer sequences used in this study were *GAPDH* 5′-ACTGGCATGGCCTTCCGT-3′, 5′-CCACCCTGTTGCTGTAGCC-3′, *IL-1β* 5′-ACAAAATACCTGTGGCCTTGG-3′, 5′-GGTGCTGATGTACCAGTTGGG-3′, *IL-6* 5′-GATGGATGCTTCCAATCTGGA- 3′, and 5′-AGCCACTGGTTCTGTGCCT-3.

#### 4.2.5. NCP Treatment Methods on Cells

This experiment was conducted to determine whether the anti-inflammatory effect of NCP acts directly on cells or depends on the components of the medium. NCP treatment was performed in three ways as shown in Figure 2A. First, in the direct method (DT): HGFs were seeded in a 35 mm dish according to the conventional method, and a distance of 1 cm was maintained between the cells and NCP device for 5 min in a 2 mL medium containing PG-LPS at a concentration of 1 μg/mL. In the DT+MC method, the medium was changed after DT treatment. The medium was removed immediately after NCP treatment, the cells were washed with PBS, and a new PG-LPS medium was added to the cells. This method minimizes the influence of the medium. In the indirect treatment (IDT) method, cells are treated by first treating them with NCP in a PG-LPS medium without cells. This method transfers only the characteristics of the medium denatured by NCP to the cells.

#### 4.2.6. NCP Treatment in the Presence of Two Types of Mesh

This experiment was conducted to determine whether there was a difference in anti-inflammatory efficacy depending on whether particles with the same charge as electrons emitted when plasma was generated passed through them. A method of removing charged particles was used using a metallic mesh.

As shown in Figure 3A, the two types of meshes used in the study were each made of cloth (dielectric, DE) and copper with the same aperture ratio and were made in the size of the lid of a 35 mm culture dish. The connected wires were grounded for the electrically grounded (EG) copper mesh. Each of the corresponding meshes was treated with NCP for 5 min while placed on a cell culture dish during each NCP treatment.

### 4.3. In Vivo Study

#### 4.3.1. Periodontitis Animal Model

The periodontitis model used in this study was partially modified using the experimental method described by Leira et al. [31]. Five-week-old male SD rats purchased from KOATECH were fed a standard laboratory diet in an animal laboratory, with a temperature of 18–25 °C being maintained and a 7-day adaptation period before the experiment.

As shown in Figure 4A, rats were divided into three groups: control (injected only with PBS) and experimental groups (injected only with PG-LPS, treated with NCP combined with PG-LPS). Rats were anesthetized by intraperitoneal injection of a cocktail of ketamine (80 mg/kg) and xylazine (8 mg/kg). Then, 3 μL of PBS or 3 μL of PG-LPSs (final concentration of 10 μg/μL) was injected into the gums between the first and second molars, and NCP treatment was performed for 5 min. The experiment was performed for 2 weeks. After 24 h from the last treatment, rats were sacrificed using CO_2_ gas in a chamber. The mandibles were collected to confirm changes in the mandible and periodontal tissue. Experiments were performed according to relevant guidelines, protocols, and regulations of the Pusan National University Institutional Animal Care and Use Committee (PNU-2020-2521).

#### 4.3.2. Micro-CT

The mandibular tissues were scanned using InspeXio SMX-90CT (Shimadzu, Gyoto, Japan) a system equipped with a 0.5 mm aluminum filter, 15% beam hardening correction, and reduction of ring artifacts. Each tissue was photographed, scanned at 0.7°, and saved in TIFF format with 19.7 μm resolution. The data scanned by micro-CT were used to reconstruct two-dimensional slice images around the teeth and alveolar bone using BON-FCS to form a three-dimensional structure, and tissue changes were observed in all directions and widths. The degree of alveolar bone resorption and changes in the periodontal ligament were confirmed using photographs.

#### 4.3.3. RNA Extraction and RT-PCR

Part of the periodontal tissue was collected from the mandible of the rats, immersed in TRIzol reagent (Invitrogen, Waltham, MA, USA), and ground as much as possible with a tissue grinder. The RNA extraction process was the same as that in the in vitro process. RNA was synthesized into cDNA using Maxime RT Premix, followed by RT-PCR with denaturation at 95 °C for 30 s, annealing for 30 s, and extension at 72 °C for 60 s. The primer sequence used in this study was *GAPDH* 5′-CCTGGAGAAACCTGCCAA-3′, 5′-GGCCATGTAGGCCATGAG-3′, *IL-1β* 5′-TCCCTGTGGCCTTGGGCC-3′, 5′-GGAAGACACGGGTTCCATGG-3′, *TNFα* 5′-CTGCCCCGACTATGTGCTC-3′, and 5′-ACCTGCCCGGACTCCGTG-3′.

#### 4.3.4. Hematoxylin and Eosin (H&E) Staining

Rat mandibular tissues were fixed in 4% PFA solution and then decalcified with 10% EDTA solution; next, the fabricated paraffin block was cut to a thickness of 5 μm. The blocks were deparaffinized, hydrolyzed, washed with water, stained with hematoxylin and eosin, dehydrated, transparentized, and encapsulated. The changes in the number of cells in the periodontal ligament were confirmed using an optical microscope (Axioscope 5; ZEISS, Jena, Germany).

#### 4.3.5. Immunohistochemistry

The paraffin sections were washed in xylene for 5 min each to remove the paraffin. The slides were washed in 100% ethanol for 5 min each to remove xylene in a hydrolysis process. After that, tissue sections were washed in PBS to remove the ethanol component completely. HRP/diaminobenzidine (ABC) detection immunohistochemistry kits were used.

Hydrated tissue sections were treated with hydrogen peroxide blocking solution for 5 min at room temperature and then washed with PBS. Then, tissues were treated with a protein-blocking solution for 5 min at room temperature and washed with PBS. After washing, CD4 and CD68 primary antibodies were diluted according to the concentration used. The tissue was submerged in the diluted antibody solution, placed in a humid box, incubated overnight at 4 °C, and washed with PBS. After treatment with biotinylated goat anti-polyvalent antibodies for 10 min, the sections were washed with PBS. Furthermore, diaminobenzidine was developed appropriately, and counterstaining was performed with Mayer’s hematoxylin for 5 min. After the dyeing and dehydration process, the samples were washed in xylene to remove the ethanol component. Changes in tissues were observed after encapsulation.

#### 4.3.6. Tartrate-Resistant Acid Phosphatase (TRAP) Assay

The TRAP assay was performed to identify osteoclasts using a solution prepared according to the method described by Cole et al. [32]. After adding 4 mg of naphthol AS-BI phosphate as a substrate in 0.2 M acetate buffer, 35 mg of fast red violet LB was added as a coupler. Then, 60 μL of magnesium chloride and 187.5 mg of L (+)-tartaric acid were added such that the final concentration was 50 mM. Finally, the pH was adjusted to 5.0 using sodium hydroxide. The reaction solution prepared in this way was applied to the tissue from which paraffin was removed, incubated at 37 °C for 1 h, washed with PBS, and counterstained with hematoxylin. After the dyed tissue underwent encapsulation, changes in the osteoclasts were confirmed.

### 4.4. Ethical Statement

This study was performed according to ARRIVE guidelines. All experimental protocols were approved by a Pusan National University Institutional Animal Care and Use Committee (PNU-2020-2521).

### 4.5. Statistical Analysis

All experiments were repeated at least three times. The data were displayed as mean ± standard deviation (SD), and statistical analysis was performed using SPSS software version 29.0.1 (Chicago, USA). Multiple comparisons were made using one-way ANOVA followed by a Tukey’s test. A two-tailed *p* < 0.05 was considered statistically significant.

## Figures and Tables

**Figure 1 ijms-25-06161-f001:**
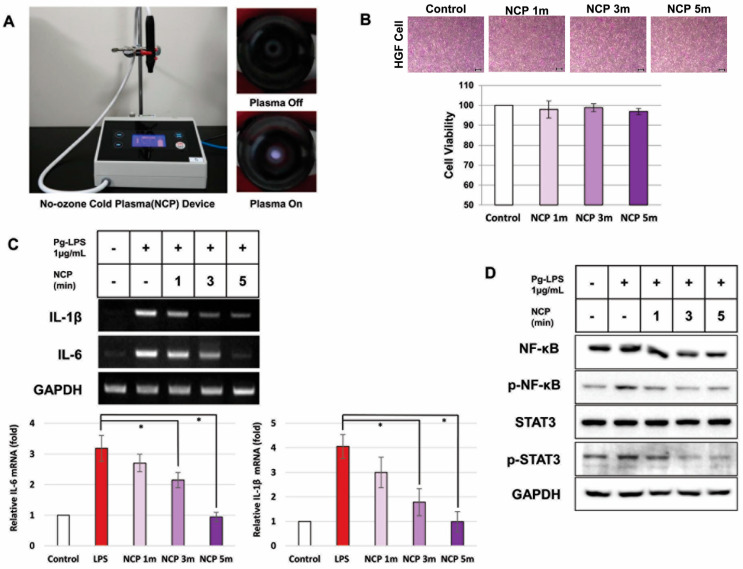
Anti-inflammatory effect according to cell viability and treatment time for the NCP device. (**A**) NCP device used in this study. When NCP is generated, purple light can be observed, as shown in the picture. (**B**) Cell morphology and viability when HGFs are treated with NCP over time. The cells were treated for 1, 3, and 5 min. (100X) Scale bar size is 20 μm. (**C**) Changes in pro-inflammatory cytokines in HGFs treated with PG-LPS and then with NCP. In the group treated only with PG-LPS, *IL-1β* and *IL-6* increased 3.1-fold and 4-fold compared with the control group. (**D**) Changes in proteins involved in the inflammatory response when inducing inflammation in HGFs with PG-LPS and then treating them with NCP. (* *p* < 0.05).

**Figure 2 ijms-25-06161-f002:**
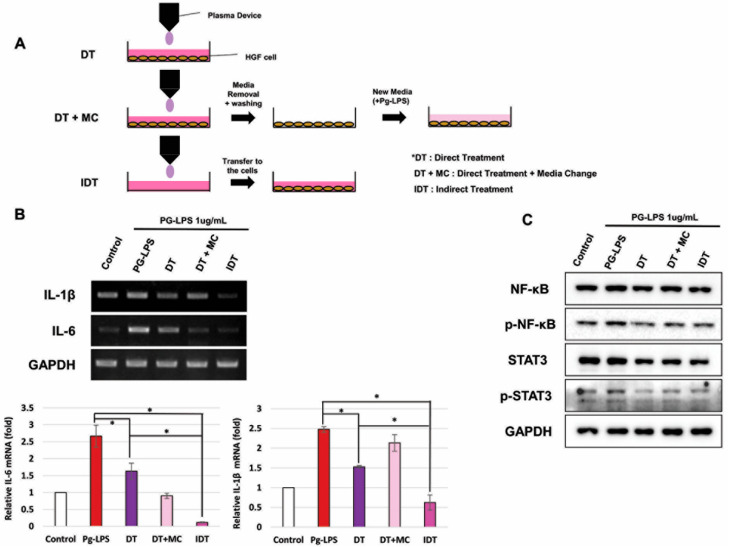
The anti-inflammatory effect was compared by treating HGFs with NCP in three ways. (**A**) Scheme of the process of treating HGFs with NCP in three ways. (**B**) Changes in pro-inflammatory cytokines according to the NCP treatment method (* *p* < 0.05). (**C**) Changes in proteins involved in the inflammatory response according to the NCP treatment method.

**Figure 3 ijms-25-06161-f003:**
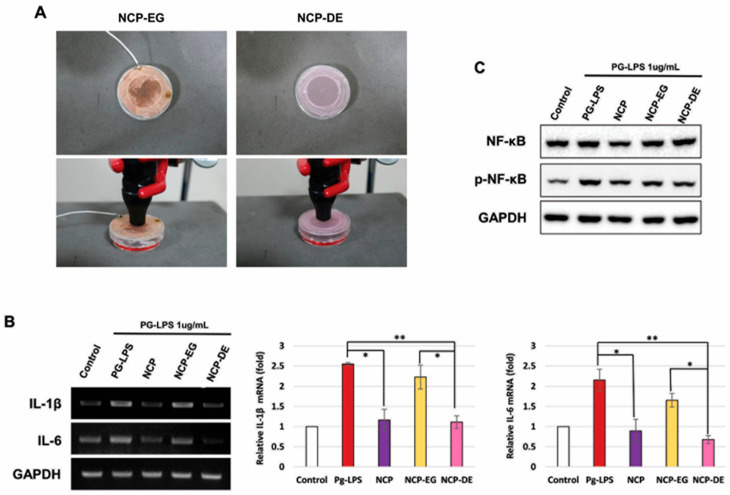
Before NCP treatment, after covering the metal/non-metal mesh, the inflammatory response was induced with PG-LPS, and then the anti-inflammatory effect was compared. (**A**) Photograph of processing NCP by applying the metal/non-metal mesh used in the experiment to actual cells. (EG: Electric ground, DE: Dielectric) (**B**) Changes in pro-inflammatory cytokines according to mesh materials (* *p* < 0.05, ** *p* < 0.01). (**C**) Changes in NF-κB according to mesh materials.

**Figure 4 ijms-25-06161-f004:**
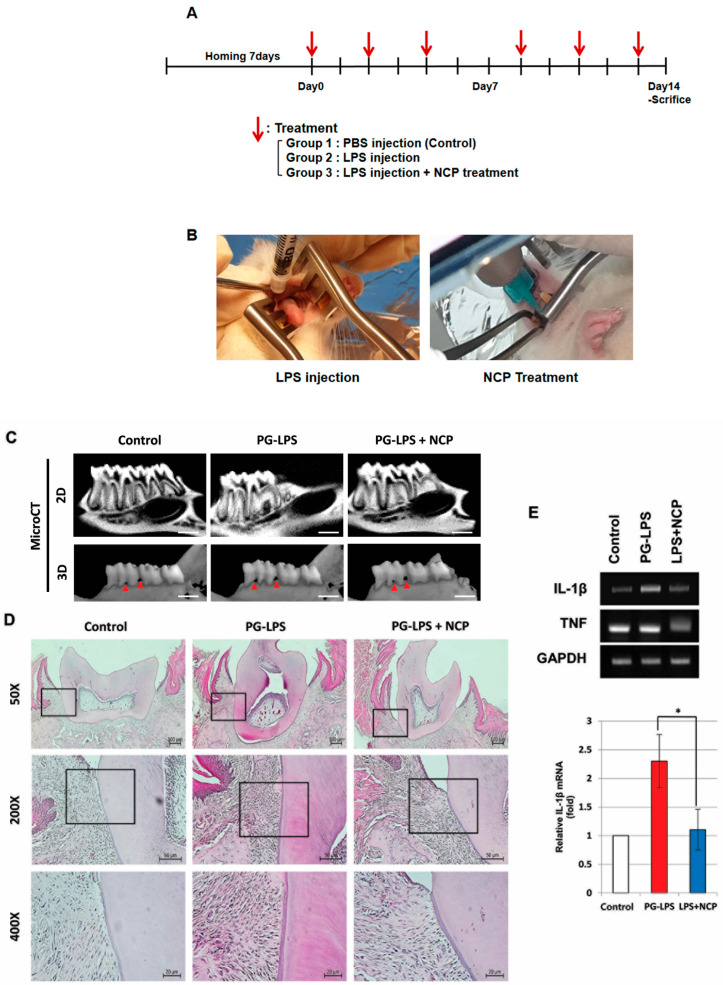
After inducing periodontitis by injecting PG-LPS into the periodontium between the first and second molars of a rat, the anti-inflammatory effect was confirmed by treatment with NCP. (**A**) The timeline of the in vivo study. (**B**) After PG-LPS injection between the rat’s mandibular first and second molar, NCP was applied at the same location for 5 min. (**C**) Changes in rat mandible on micro-CT image. In 3D image, the red arrow shows the space between the tooth and periodontium. Scale bar is 5 mm. (**D**) Changes in the cell number in periodontal tissues near the teeth confirmed by H&E staining. (**E**) Pro-inflammatory cytokine changes confirmed by RT-PCR in rat periodontal tissues. (* *p* < 0.05).

**Figure 5 ijms-25-06161-f005:**
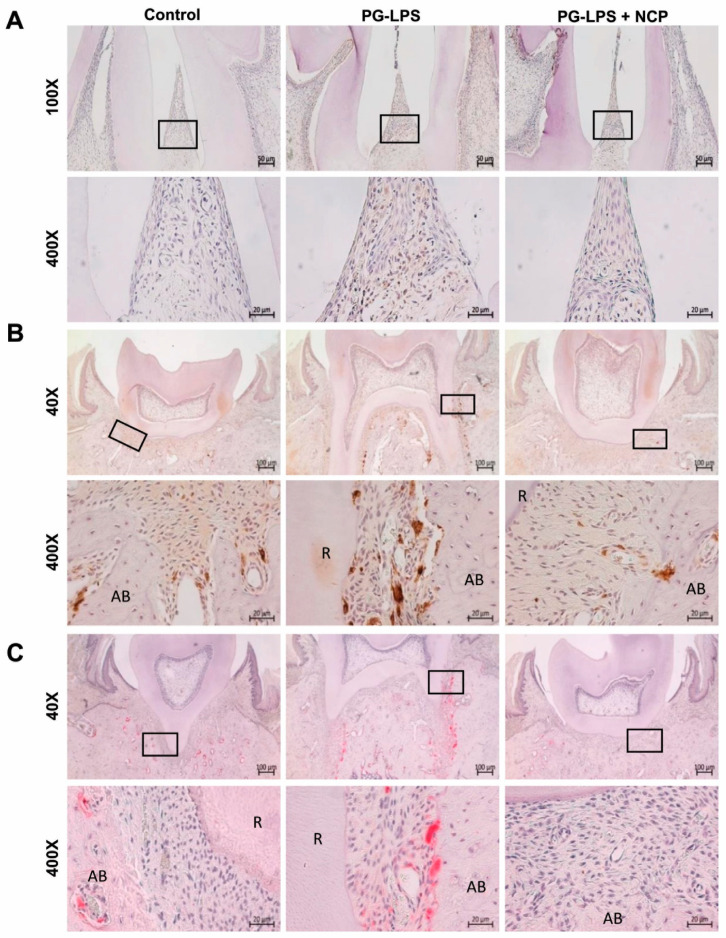
When NCP was applied to the mandible of rats with periodontitis, changes in the inflammatory response in periodontal tissue were confirmed. The black box area is enlarged below. (**A**) Changes in T cells confirmed by immunohistochemistry. CD4 was used as a marker for T cells. (**B**) Changes in macrophages confirmed by immunohistochemistry. CD68 was used as a macrophage marker. (**C**) Changes in osteoclasts confirmed by the TRAP assay. AB: alveolar bone, R: root.

## Data Availability

The datasets used and/or studied during the current study are available from the corresponding author on reasonable request.

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
