# Peer review of "Anti-Inflammatory Activity of No-Ozone Cold Plasma in Porphyromonas gingivalis Lipopolysaccharide-Induced Periodontitis Rats"

_ijms, 2024, doi:10.3390/ijms25116161_

Round 1

Reviewer 1 Report

Comments and Suggestions for Authors

The manuscript investigates the anti-inflammatory activity of No-ozone Cold Plasma (NCP) in periodontitis induced by Porphyromonas gingivalis (P. gingivalis). The study explores the effect of NCP on human gingival fibroblasts (HGF) and a rat model of periodontitis, aiming to demonstrate the potential of NCP as a safe and effective treatment for periodontitis.

1. Main Question Addressed by the Research

The main question addressed by the research is whether No-ozone Cold Plasma (NCP) can be used safely and effectively to treat periodontitis, an inflammatory disease caused by P. gingivalis, by examining its anti-inflammatory effects in vitro on HGF cells and in vivo in a rat model.

2. Originality and Relevance

-Originality The use of NCP to treat periodontitis is novel, especially given the focus on avoiding ozone generation, which poses respiratory risks.

-Relevance: The study addresses a significant gap in periodontal treatment by offering a potentially safer alternative to conventional methods that often involve thermal damage or residual microorganisms causing inflammation.

3. Contribution to the Subject Area

The study contributes to the subject area by:

- Introducing a non-thermal, ozone-free plasma treatment for periodontitis.

- Providing evidence of the anti-inflammatory effects of NCP on P. gingivalis-induced periodontitis.

- Highlighting the potential mechanism of action of NCP, including its impact on inflammatory cytokines and proteins.

 4. Methodological Improvements

Improvements Suggested:**

  - Comparisons with other standard treatments like antibiotics or scaling.

-Methodological Details: More detailed descriptions of the NCP device and its operation parameters would be beneficial.

- Statistical Analysis: Ensure robust statistical analyses to validate the results, including more replicates to increase the reliability of the findings.

5. Consistency of Conclusions with Evidence

The conclusions are generally consistent with the presented evidence. The study shows that NCP reduces inflammatory responses both in vitro and in vivo, supporting its potential as a treatment for periodontitis. The experiments conducted, such as measuring cytokine levels and protein activities, effectively address the main research questions.

Points of Improvement:**

- Discuss potential long-term effects of NCP treatment not covered in the study.

6. Appropriateness of References

The references are appropriate and relevant, citing recent and foundational studies on periodontitis, plasma technology, and inflammatory mechanisms. Ensuring the inclusion of the most recent studies would strengthen the manuscript.

 7. Additional Comments on Figures and Data Quality

Figures:

- Ensure all figures are clearly labeled and include detailed captions. What is NT in figures?

- Improve the readability of graphs and images by using larger fonts and high-resolution images.

Data Quality:

- The data appears to be of high quality, with thorough experimental procedures and controls.

Major Issues

1. Methodology: The study needs more detailed descriptions of the NCP treatment protocols and additional control groups.

2. Statistical Analysis: Increase the number of replicates and ensure robust statistical validation of results.

Minor issues

Correct some errors in the text, such as the reference on line 210.

Conclusion

The manuscript provides a promising new approach to treating periodontitis using NCP. Addressing the methodological and minor issues will enhance the robustness and clarity of the study, supporting its potential impact on periodontal therapy.

Author Response

Thank you for reviewing my paper. I would like to supplement and respond to a few points.

- Comparisons with other standard treatments like antibiotics or scaling.

: Thank you for your good comments. Currently, a dental treatment device has been developed based on this plasma device, and exploratory clinical tests have been completed. Through future clinical tests, it is expected that we will be able to more clearly compare the differences with other treatment methods such as antibiotics or scaling, and we hope to report this in a paper.

- Discuss potential long-term effects of NCP treatment not covered in the study.

: Our research team has continued research in the field of plasma medicine using our proprietary plasma device. Non-clinical and preclinical tests were conducted in various fields such as atopic dermatitis, anti-cancer, anti-inflammatory, and wound healing, and showed meaningful results. Based on these results, we are developing medical devices that apply NCP in actual clinical trials, and we believe that if we conduct clinical trials using the developed medical devices, we will be able to confirm the long-term effects of NCP. As short-term preclinical results show that NCP treatment improves the condition without side effects, it is expected that long-term treatment with NCP will also continue to alleviate the condition. In particular, in areas that require long-term treatment, such as skin regeneration and anti-cancer treatment, it is expected that combining NCP with existing treatment can produce effects such as shortening the treatment period.

-The references are appropriate and relevant, citing recent and foundational studies on periodontitis, plasma technology, and inflammatory mechanisms. Ensuring the inclusion of the most recent studies would strengthen the manuscript.

: Some references were re-searched and revised.

- Ensure all figures are clearly labeled and include detailed captions. What is NT in figures?

: NT is an abbreviation for non-treated and has the same meaning as control. To prevent confusion, all NTs in the figure have been changed to control.

- Improve the readability of graphs and images by using larger fonts and high-resolution images.

: The image quality has been improved to 600 dpi and some fonts have been significantly modified.

  1. Methodology: The study needs more detailed descriptions of the NCP treatment protocols and additional control groups.
  2. Statistical Analysis: Increase the number of replicates and ensure robust statistical validation of results.

: We have supplemented the content in this regard.

Correct some errors in the text, such as the reference on line 210.

: It was corrected after confirmation.

Reviewer 2 Report

Comments and Suggestions for Authors

Dear authors, thank you very much for this interesting report, which certainly shows a new and promising approach for the treatment of periodontitis. The aim of this preclinical study was to investigate the effects of No-ozone cold plasma (NCP) in vitro on human gingival fibroblasts (HGF) treated with P. gingivalis-derived lipopolysaccharide (PG-LPS) and in vivo in rats suffering from a periodontitis-like inflammatory response.

To improve your manuscript, I suggest the following changes that should be made by the authors:

- I see technical errors in the text, such as unnecessary “spaces" through the main text, the parenthesis with the reference should come before the period, etc… (e.g. line 31, 38 etc…....)

- The introductory section provided a good insight into the known facts describing the ethiopathogenesis of periodontitis and the possible treatment options. In addition, the authors provided a logical rationale for their hypothesis formulation and the formulation of the research objectives.

- The M&M section is very well described, but I would like to know why you did not also study prolonged application of NCP to HGF in rats (i.e. 8, 10 minutes?).

- Please add the number of the ethical statement approval

- Figure 5, please add what AB and R stand for in the figures.

 I look forward to your reply. Kind regards!

Comments on the Quality of English Language

No comments

Author Response

Thank you for reviewing my paper. I would like to supplement and respond to a few points.

- I see technical errors in the text, such as unnecessary “spaces" through the main text, the parenthesis with the reference should come before the period, etc… (e.g. line 31, 38 etc…....)

: It was corrected after confirmation.

- The M&M section is very well described, but I would like to know why you did not also study prolonged application of NCP to HGF in rats (i.e. 8, 10 minutes?).

: Our research team aims to develop a dental treatment device based on the periodontitis treatment effect of NCP. When initially designing the experimental protocol, we investigated the time the device could be applied to patients in actual clinical practice and found that the maximum was 5 minutes. We judged that it would be difficult for both the patient and the operator if the procedure time exceeded 5 minutes when using the device inside the oral cavity, so we did not allow it to exceed 5 minutes in actual in vitro and in vivo experiments.

  Reflecting this, as shown in Fig. 1, the plasma treatment time was set to 1 minute, 3 minutes, and 5 minutes in the safety test of the device. In addition, it can be seen that the anti-inflammatory effect also improved as the treatment time increased under the same conditions. In particular, it can be seen that the maximum processing time of 5 minutes showed anti-inflammatory efficacy that was not significantly different from the control.

Based on the actual clinical device application time and in vitro experiment results, it was concluded that 5 minutes of plasma treatment time was sufficient, and subsequent experiments were conducted.

- Please add the number of the ethical statement approval

: The number of Ethical statements has been added. (line 413)

- Figure 5, please add what AB and R stand for in the figures.

: The meaning of AB and R has been added to the end of the description of Figure 5. (line 197) 

Round 2

Reviewer 1 Report

Comments and Suggestions for Authors

Once the changes have been made and following the advice of this reviewer, the article can be published in this journal

Reviewer 2 Report

Comments and Suggestions for Authors

Dear Authors, thank you for your kind answers! Wish you all the best!

Comments on the Quality of English Language

No comments